# Quantifying the availability of seasonal surface water and identifying the drivers of change within tropical forests in Cambodia

Louisa Mamalis[1,2]*, Kathryn E. Arnold[3], Simon P. Mahood[4], Mao Khean[5], Colin M. Beale[1,2,6]

1 Department of Biology, University of York, York, United Kingdom, 2 Leverhulme Centre for Anthropocene Biodiversity, University of York, York, United Kingdom, 3 Department of Environment and Geography, University of York, York, United Kingdom, 4 Wildlife Conservation Society, Gasabo, Kigali, Rwanda, 5 Wildlife Conservation Society, Sangkat Tonle Bassac, Khan Chamkarmorn, Phnom Penh, Cambodia, 6 York Environmental Sustainability Institute, University of York, York, United Kingdom

* louisa.mamalis@york.ac.uk

**Data Availability Statement:** The study utilised existing data that are publicly available from Google Earth Engine https://developers.google.com/earth-engine/datasets/ and from Open Development

## Abstract

Surface freshwater is a vital resource that is declining globally, predominantly due to climate and land use changes. Cambodia is no exception and the loss threatens many species, such as the giant ibis a Critically Endangered waterbird. We aimed to quantify the spatial and temporal (2000–2020) change of surface water availability across northern and eastern Cambodia and to assess the impact of this on the giant ibis. We used a Random Forest Classifier to determine the changes and we tested the impact of land use and geographical covariates using spatially explicit regression models. We found an overall reduction of surface water availability of 4.16%. This was predominantly driven by the presence of Economic Land Concessions and roads which increased the probability of extreme drying and flooding events. The presence of protected areas reduced these probabilities. We found changes in precipitation patterns over the wider landscape did not correlate with changes in surface water availability, supporting the overriding influence of land use change. 98% of giant ibis nests recorded during the time period were found within 25m of surface water during the dry season, highlighting their dependency on surface water. The overall surface water decline resulted in a 25% reduction in dry season suitable habitat for the giant ibis. Although absolute changes in surface water over the whole area were relatively small, the impact on the highest quality habitat for ibis is disproportionate and therefore threatens its populations. Defining the threats to such an endangered species is crucial for effective management.

## Introduction

Freshwater availability is increasingly affected by climate change and human activities [1]. Land use changes such as deforestation, mining and hydroelectric dams negatively impact surface water availability [1, 2]. Climate change is an increasing threat to the provision of surface water and the predicted changes rainfall patterns and increasing temperatures will lead to

Cambodia https://data.opendevelopmentcambodia.net/map-explorer. We have also included the Google Earth Engine code for creating the surface water classifier: https://code.earthengine.google.com/3befd45f9aa16ceba3ad4fb6783ef765 We have included the code for the creation of the surface water transition maps: https://code.earthengine.google.com/aee70dbbf330de45410ecb54aaffa687 We have also included the code for computing the distance to surface water of the giant ibis nests and the non-nest points: https://code.earthengine.google.com/a23b2bbe5b027f106fa859dbbb55fef5 The code for computing and summarising the distance to surface water from giant ibis nest points and non-nest points is included here: https://colab.research.google.com/drive/1DV54sr475YpnIV7lTMM6-cGnINm56rZg. All data and scripts used for the the analysis within this research can be found at the following github repository; https://github.com/LouMamalis/Quantifying-availability-of-surface-water-change-within-tropical-forests-in-Cambodia However, the data regarding the locations of the giant ibis nests will not be shared as that is of sensitive nature and needs to remain confidential so as not to harm the persistence of the species.

**Funding:** This work was funded by the Natural Environment Research Council, UK Research Institute under the Adapting to the Challenges of a Changing Environment DTP https://accedtp.ac.uk/. The grant was awarded to LM to complete this work [grant code: NE/L002450/1, 2020]. the funders had no role in study design, data collection and analysis, decision to publish, or preparation of the manuscript.

**Competing interests:** The authors have declared that no competing interests exist.

increased droughts, loss of water sources as well as extreme flooding in other areas [3, 4]. These change are having a profound effect on species dependent on freshwater, particularly those within tropical regions that experience extreme seasonal water scarcity [5, 6]. Animals rely on surface water for drinking, food resources and thermoregulation so access is vital and influences species behaviour and distribution [7]. For example, water within the Serengeti plains drives the mass migration of ungulates [8], and waterholes are important congregation points for peccaries in Guatemala during the dry months [9]. Humans also rely on surface water using it for drinking, agriculture, livestock, provision of energy and transport [10]. Lack of freshwater is a serious problem, with ~4 billion people already living in water scarce regions, which is likely to increase alongside affluence and water consumption [11]. As global affluence increases, so too will water consumption, making the continued provision of sufficient water a serious challenge [12].

Globally, freshwater systems are under studied and underrepresented within policy despite their importance for people and biodiversity [13]. For example, over half the world's wetlands occur in the tropics, with a large number of people and animals relying on these resources, yet there is little literature about changes to surface water in these systems [14]. This is particularly true for Cambodia which is lacking data on surface water changes, especially at a large spatial scale with those that are available focussing mainly on the Tonle Sap lake system [14–16]. Cambodia is located within the Lower Mekong Basin meaning it has extensive seasonal surface water which is vital for its biodiversity and people [17]. Surface water here forms numerous rivers and the Tonle Sap, the largest lake in southeast Asia, creating important habitats for species [18]. There is a mosaic of floodplain, wetland, swamp and lagoon habitats which fluctuate dramatically with the seasonal rains, transforming the landscape in the wet season [19]. Seasonal waterholes are also important, providing refuge for many threatened species during the dry season when freshwater is scarce [20]. Surface water is also valued by people, providing freshwater and fishing resources, with 80% of the population relying on agriculture and thus freshwater [16, 21]. Surface water forms a vital part of Cambodia's landscape, for both humans and animals, therefore it is vital to understand how climate and land use changes are affecting it.

In light of land use and future climate change threats to this system we aim to quantify how surface water has changed in northern Cambodia using remote sensing which allows us to explore changes at a large spatial scale [1]. Advances in satellite imagery and remote sensing methods have begun to enable detailed assessments of land use and surface water changes on a large scale [22, 23]. However, how changes in surface water impact habitat availability for specific species is rarely evaluated. We therefore couple our assessment of surface water change with an evaluation of how the changes impact habitat availability for a critically endangered bird, the giant ibis *Thaumatibis gigantea*. We chose the Cambodian national bird as an exemplar species due to its reliance on surface water in order to highlight the need to understand how hydrological systems are changing. We formed three hypotheses; 1) the availability of surface water within northern and eastern Cambodia has reduced since 2000, 2) the drivers of the changes in surface water availability would be human land use changes, such as large-scale agriculture and roads and 3) the reduction in availability of surface water will have a negative impact on the availability of suitable nesting habitat for the giant ibis.

## Materials and methods

### Study area

The study was conducted within the estimated range of the giant ibis, across the northern and eastern plains of Cambodia (13.62˚, 105.68˚) within the Indo-Burma biodiversity hotspot (Fig 1). The climate within this region is governed by distinct wet (Dec-May) and dry (Apr-Nov)

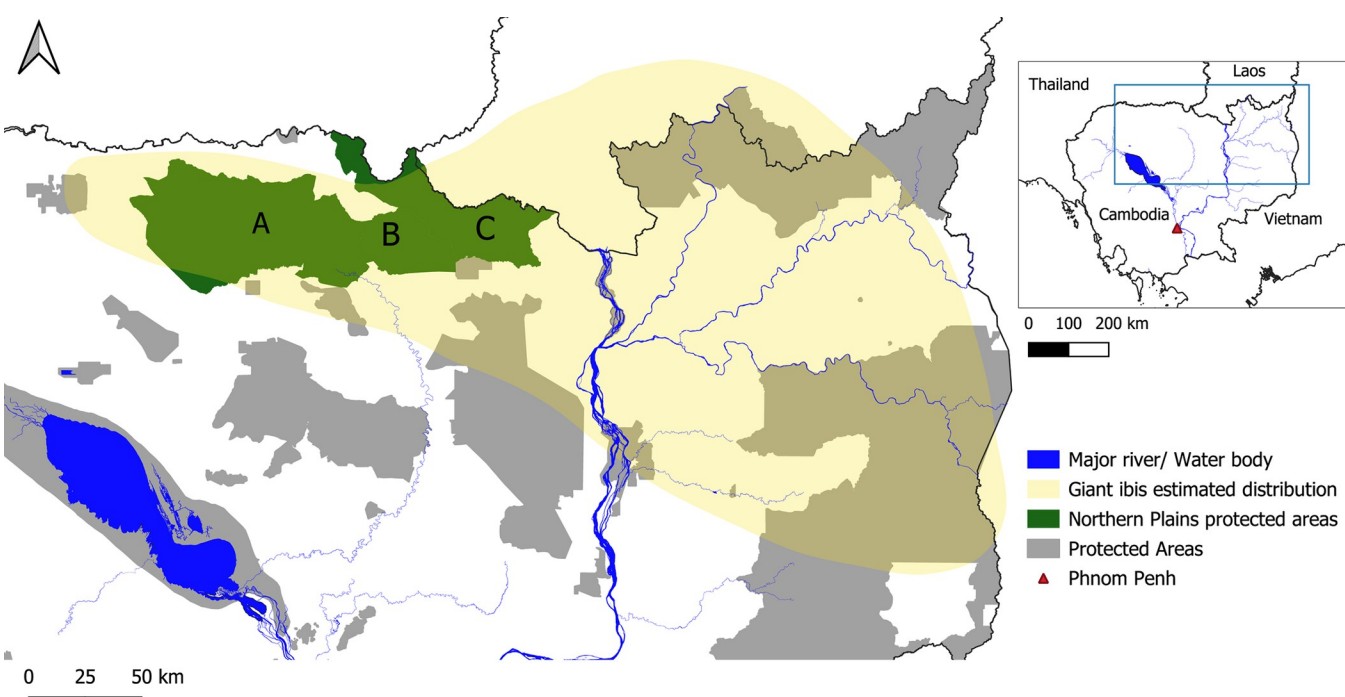

**Fig 1. Map to show the study site.** This map shows the study site which is the estimated giant ibis distribution across the north east of Cambodia (yellow envelope) [34]. The grey polygons show all the protected areas across Cambodia and the protected areas we have giant ibis nest data are highlighted in dark green; A) Kulen Promtep Wildlife Sanctuary, B) Prey Preah Roka Wildlife Sanctuary and C) Chhaeb Wildlife Sanctuary. The country boundary data is reprinted from geoBoundaries under a Creative Commons Attribution 4.0 International licence. Protected areas and water bodies data included in this figure has been published by Open Development Cambodia herein are licensed under a CC BY-SA 4.0. The giant ibis distribution data included in this figure has been reprinted from [34] under a CC BY licence, with permission from BirdLife International, original copyright [2019].

seasons driven by the tropical monsoons [24]. The mean annual rainfall across the central low-land regions is 1400mm, with average temperatures of 28˚C and the highest temperatures of 38˚C degrees recorded in April [24, 25]. This landscape is made up of a mosaic of deciduous dipterocarp forest, semi-evergreen forest, evergreen forest, areas of seasonally-flooded grasslands, bamboo forests, seasonally-flooded riparian habitats and a network of temporary and permanent forest pools and streams [26, 27]. Deciduous dipterocarp forest is the predominant habitat type and it is an endemic community of south and southeast Asia, characterised by sparse tree cover and a dense understory of grasses studded with waterholes [21, 28]. It is disappearing rapidly due to conversion for large-scale areas of cash crops such as rubber and cassava as well as expansion of small holdings, settlements, roads, mining and logging [28–31]. These large-scale areas of cash crops, also known as Economic Land Concessions (ELCs), are prevalent, covering nearly 8000 km$^2$ across the study site (S1 Fig).

We used giant ibis nest records as part of this study which were from Kulen Promtep Wildlife Sanctuary, Prey Preah Roka Wildlife Sanctuary and Chhaeb Wildlife Sanctuary (as of 2023 combined to form Chhaeb-Preah Roka Wildlife Sanctuary [32]), protected areas found within this region are jointly managed by the Forestry Administration of the Ministry of Agriculture, Forestry and Fisheries and the Wildlife Conservation Society [33]. These protected areas create a continuous corridor of landscape (10,796km$^2$) across the Northern Plains forming one of the largest remaining areas of deciduous dipterocarp forest [33]. These protected areas are very important for biodiversity supporting populations of at least 15 threatened species such as gaur *Bos gaurus*, banteng *Bos javanicus*, sarus crane *Antigone antigone*, and giant ibis *Thaumatibis gigantea*, making this an extremely important region for biodiversity [27].

## Giant ibis, *Thaumatibis gigantea*

For this study we applied remote sensing techniques directly to the conservation of the giant ibis, a Critically Endangered waterbird. Historically present across Thailand, Cambodia, Laos PDR and Vietnam they are now confined predominantly to Cambodia, with potentially transient individuals recorded in Vietnam and Laos PDR [35]. Its populations are small and declining, with ~200 individuals remaining across a small number of protected areas [36]. The giant ibis relies on deciduous dipterocarp forests and waterholes for forage in the shallow water and saturated mud [37]. The birds breed during the wet season, pairing and nesting between June and September [35]. The giant ibis is predominantly threatened by habitat loss; in particular of waterholes and large nesting trees, incidental poisoning at waterholes, human disturbance, natural predation of chicks in the nest and to a lesser degree hunting [38, 39]. This species has been the focus of conservation work by the Ministry of Environment, local and international conservation organisations but the populations are not increasing, creating the need for further study [40].

## Data analysis

**Temporal surface water change.** To evaluate changes in surface water over time, we created a surface water classifier using Landsat 7 satellite images and a water index in Google Earth Engine (GEE) [41]. We chose to use Landsat 7 images because the images have a 30m resolution and have been taken every 16 days for the last 30 years which provides extensive detailed and long-term spatial data for monitoring changes [1, 42]. This resolution was suitable for our study as waterholes within the landscape are rarely smaller than 30m, reducing the chances that they would be missed in analysis. Before we began the final analysis with Landsat 7 images we completed some pre-processing. We applied a cloud mask function in GEE to mask out any clouds and their shadows present in the images. The presence of clouds and cloud shadows can affect the analysis process so we created a cloud mask in GEE to remove this bias [42]. This cloud mask first uses the Pixel Quality Assurance band of the images to determine whether each pixel is affected by cloud or not [43]. We masked areas of the image identified as cloud shadow (3), cloud (5) and cloud confidence (7) and that pixels affected by cloud in any bands were masked from the analysis [42, 43]. Since 2003 a scanline error has been present in the Landsat 7 data. To correct this error, we filled in the affected pixels using a morphological mean that we calculated and applied to the empty pixels. To do this we by calculated the average value of pixel values within a 2-pixel radius square neighbourhood per empty pixel to get an approximate value for these [42].

Following methods by Fisher et al. [22] we computed a water index based on high reflectance of water in the selected bands; 'blue', 'green', 'red', 'nir', 'swir1' and 'swir2' which classified pixels as water or not. We created training data for the classifier by manually identifying 187 paired water and non-water points in a transect across the study area from a 2019 Google Earth image. 30% of this data was used for validating accuracy of the classifier [44]. We input the data into a Smile Random Forest Classifier with 25 iterations and grouped the water index images into quarterly images to reflect the seasonality within Cambodia [41]. We summarised the processed images to compute the quarterly estimates of surface water, scoring a pixel as water if at least one image within the quarter was classified as water between 2000 and 2020 [22]. To evaluate the classifier performance, we computed the sensitivity and specificity of classification in the 30% of data held out during model training.

To examine long-term trends across a spread of data, we consolidated the quarterly surface water data into two five-year periods, 2000–2004 and 2016–2020. To quantify the change in surface water we defined four surface water states based on the number of quarters where

**Table 1. Table to outline the definition of each surface water state.**

| Surface water state | Number of quarters flooded |
| --- | --- |
| Permanently flooded | A pixel that flooded for 16 or more quarters over the five-year period |
| Irregularly flooded | A pixel that was flooded for five to 15 quarters over the five-year period |
| Rarely flooded | A pixel flooded for less than five quarters over the five-year period |
| Never flooded | A pixel that was not flooded during any of the quarters over the five-year period |

This table summarises the defined surface water states which are based on the number of quarters that each pixel was flooded for during the five-year time periods.

water was present per pixel. These states were: 1) permanently flooded; 2) irregularly flooded; 3) rarely flooded; and 4) never flooded (Table 1). We analysed the changes between the time periods which resulted in 16 possible surface water transition categories (S1 Table).

**Drivers of change analysis.** We analysed the relationship between the annual precipitation and the area of surface water using a linear regression. We completed a trend analysis of the surface water change over time to determine any general patterns of precipitation change. To examine the drivers of change in surface water, we collated data on Economic Land Concessions (ELCs), protected areas (PAs), elevation and distance to roads (S1 Text). To test the influence of the covariates on surface water we used a spatial regression approach using Integrated Nested Laplace Approximation (INLA) modelling in RStudio [45]. Integrated Nested Laplace modelling (INLA) enables efficient analysis of complex spatial data within a Bayesian context [46]. It enables fitting of linear mixed models through use of an extremely efficient approximation of the Bayesian posterior based on Laplace approximations and fitting of the spatial model using Stochastic Partial Differential Equations (SPDE) [47, 48]. It allows modelling of spatial autocorrelation, which we estimated across a ~6km triangulated mesh covering the whole study site. This approach enabled the assessment of the influence of space and potentially interacting neighbours to explain variation in data that the selected covariates may not explain [46, 48]. To fit Bayesian models, we need to provide priors that make assumptions about model structure. For the fixed effects of covariates, we chose vague priors with a slight bias towards zero (a normal distribution, mean zero and precision 0.001), for the spatial random effect we tested a range of prior values before choosing a value where further adjustments up or down made negligible difference to the posterior estimates. While there aren't currently any specific goodness of fit tests specific to INLA models, we tested standard assumptions of regression type models using diagnostic plots.

We created three separate INLA models, the first model tested the effect of the covariates on any change in surface water extreme flooding and drying. For this we assumed a gaussian data distribution, a reasonable approximation for the true pattern. The data was close to a continuous distribution, using a scale of flooding or drying. For the second and third models we used subsets of the data and a binomial model to predict patterns of 'extreme drying' and 'extreme flooding', in relation to the covariates between the two five-year time periods. A significant drying event was defined as land that became drier by two or more transition categories, for example a pixel that transitioned from irregularly to never flooded. A significant flooding event was defined as land that became wetter by two or more transition categories, for example from rarely to permanently flooded.

**Implications for the giant ibis.** Next, we assessed the implications of the change in availability of surface water on the survival of the giant ibis by examining the location of giant ibis nests and the distance to surface water. We used 438 nest locations from between 2003 and 2020 within Kulen Promtep, Prey Preah Roka and Chhaeb Wildlife Sanctuaries (S2 Fig). The

nest records were collected by biodiversity teams from the Wildlife Conservation Society during annual Bird Nest Protection Programme searching and monitoring (S2 Text). To determine whether the distance from giant ibis nests and surface water was less than expected by chance we generated 876 random points for comparison. For accurate comparison the random points were generated in direct proportion to the number of real nests recorded that year. We calculated the distance to surface water for the nest and non-nest points during the wet and dry season and analysed the difference between the two groups using a Mann-Whitney U. To calculate the loss of suitable giant ibis habitat we computed the distance to the nearest surface water for each nest, then calculated the 75th percentile of this distance during both seasons. We calculated the sum of the area of suitable nesting habitat and completed a sensitivity test before selecting the 75th percentile distances to compare the changes annually and on a longer-term basis (S2 Table).

## Results

### Changes in availability of surface water

To test our first hypothesis, we used the water index classifier to compute the availability of surface water. The accuracy of our classifier over the training partition was 97% and in the test partition accuracy was 75%. We found inter-annual variation in the total area of surface water and an overall reduction (Fig 2). We found stronger variability and more frequent extreme changes in surface water availability during the dry season and this made up a small percentage (4–13%) of the total annual surface water. The total surface water area declined from 33,9761km$^2$ (2000–2004) to 29,0825 km$^2$ (2016–2020), which equates to a 14% decline. Our trend analysis identified a slight found an overall decline in mean annual precipitation (S3 Fig) but this was not a significant decline ($F_{1,19}$ = 1.87, p > 0.05) (S3 Table). Precipitation patterns did not seem to be the driving forces for this change as we found no correlation between annual precipitation and surface water availability annually (r = -0.16, df = 19, p > 0.05) or during the dry season (r = -0.38, df = 19, p > 0.05) (S4 Table).

The majority of the land area between 2000–2004 consisted of irregularly and rarely flooded pixels (Fig 3A). These two surface water states accounted for 85% (2000–2004) of the total area and 81% (2016–2020). Pixels that were never flooded made up the next highest category, 14% of the total surface area (2000–2004) and 18% (2016–2020). Permanently flooded pixels made up the smallest percentages of both the first and last periods, 1.1% and 1.3% respectively.

Generally, we found a decline in surface water availability over time, highlighted by an increase in the drying transition categories (Fig 3B). We found that 28% of pixel transitions were from 'irregularly flooded' to 'rarely flooded', driving the decline in surface water. 9% of the pixel transitions were from 'rarely flooded' to 'never flooded', supporting the overall surface water decline. Many pixels remained 'irregularly flooded' and 'rarely flooded', 15% and 30% respectively and 6% remained 'never flooded'. Pixels classed as permanently, rarely or never flooded, showed an increase of 0.2%, 11% and 4% respectively. This resulted in 7,779km$^2$ gain of rarely and never flooded pixels driving the overall decline in surface water.

### Effect of land use change on surface water availability

The first spatially explicit regression model tested for both flooding and drying and we found that change in surface water was significantly correlated with all covariates. However, these correlations were weak and no covariate had a strong effect on change (S5A Table). Drying was slightly more common on land further from roads and on land at higher elevations. The probability of more extreme flooding events was marginally higher in ELCs and protected areas.

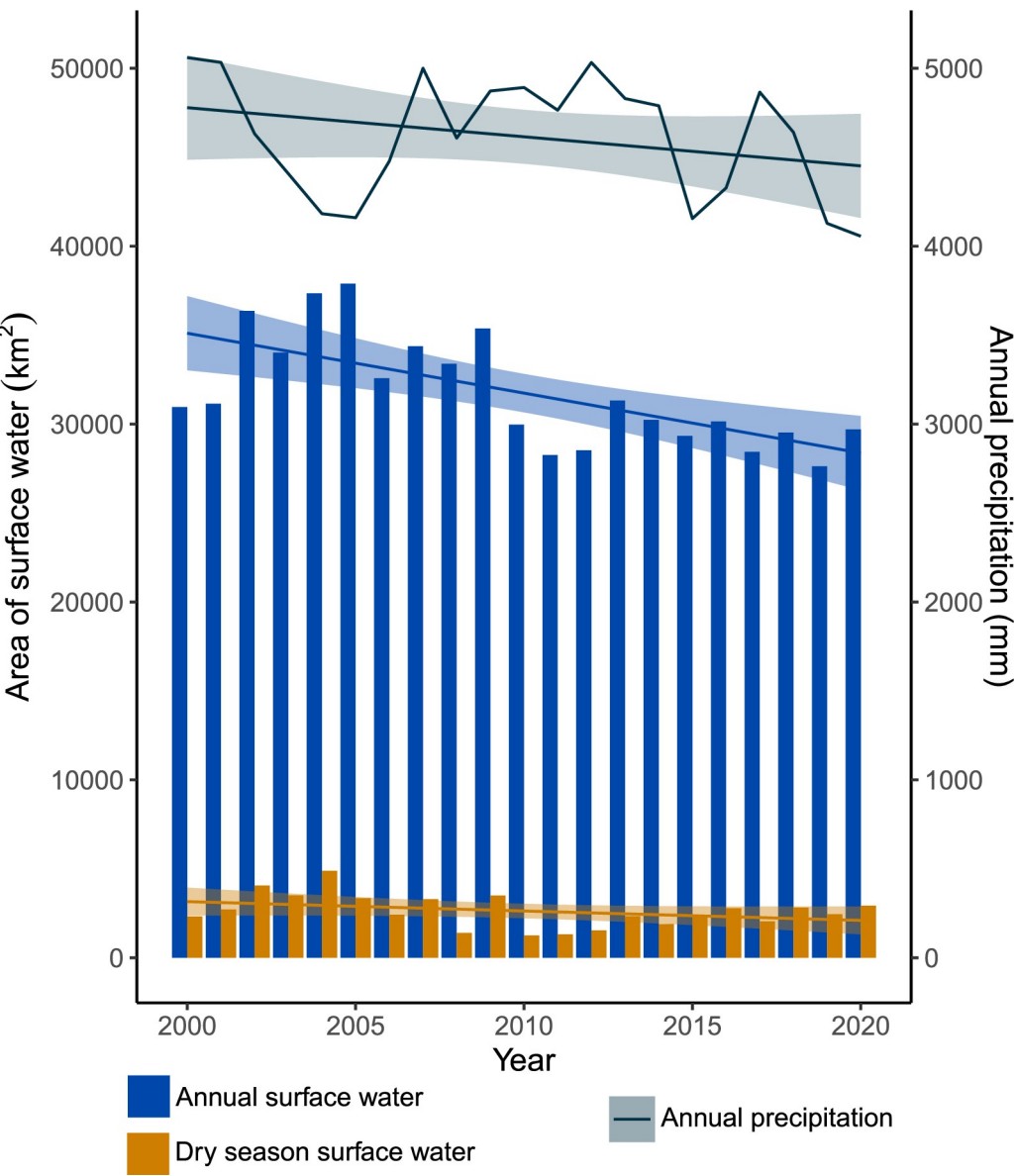

**Fig 2. Temporal plot of changes in surface water area.** This plot shows the total annual surface water area (blue bars) and during the dry season (orange bars) compared with the annual precipitation (line plot). Error lines show the 95% confidence level interval for predictions from a linear model.

Our second model tested associations of extreme drying events and the covariates. We found that change in surface water was significantly correlated with all covariates, with slightly larger effects than model one (S5B Table). We found that extreme drying events were more frequent at higher elevation and closer to roads (Fig 4A). We found that pixels within ELCs were more likely to experience extreme drying and the opposite for those within protected areas (Fig 4C).

Our third model assessed the probability of extreme flooding. Extreme flooding was significantly correlated with the presence of protected areas, which were less likely to experience extreme flooding, while ELCs had a higher probability of extreme flooding (Fig 4D and S5C Table). There was no significant correlation with the presence of roads on the probability of land flooding (Fig 4B).

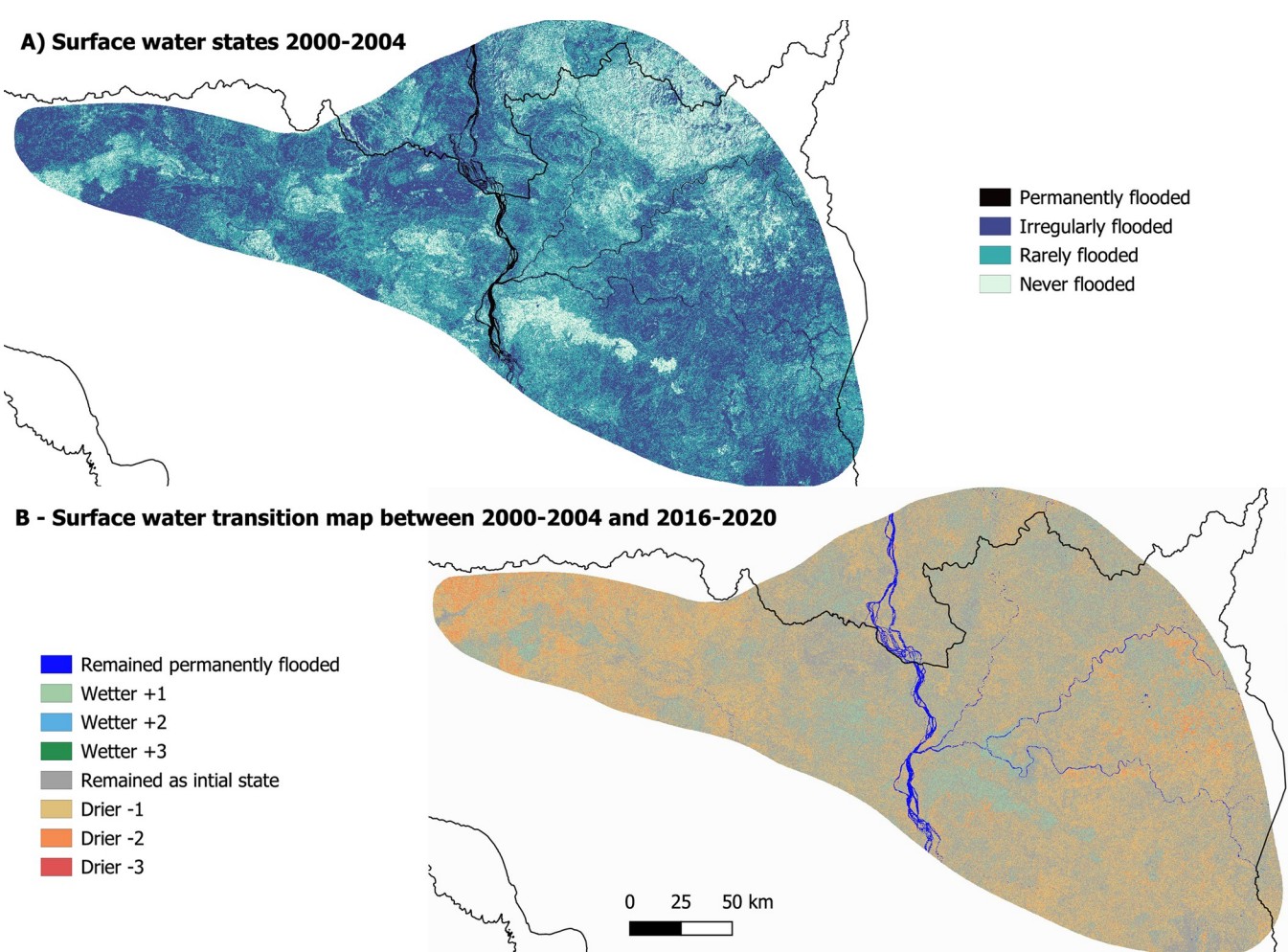

**Fig 3. Maps to show the changes in surface water availability across the study site.** A) Map of pixel surface water states between 2000 and 2004; B) Map of changes in availability of surface water between the two time periods, 2000–2004 and 2016–2020. Country boundary data is reprinted from geoBoundaries under a Creative Commons Attribution 4.0 International licence. Water bodies data included in this figure has been published by Open Development Cambodia herein are licensed under a CC BY-SA 4.0. The giant ibis distribution data included in this figure has copyright to BirdLife International and permission has been granted to the authors and PLOS ONE to publish it in this paper.

## Implications of surface water loss on giant ibis populations

Giant ibis nests were generally found closer to areas of surface water than random points during the wet (Mann Whitney U, W = 221225, N = 1314, P < 0.05) and dry season (Mann Whitney U, W = 237358, N = 1314, P < 0.05) (S6 Table). We found that 97% of the giant ibis nest points and 91% of the non-nest points were located within 25m of surface water during the wet season (Fig 5A and 5B). During the dry season both nest and non-nest points were found further from surface water, but still 42% of the nest points were found within 25m of surface water, compared with 28% of the non-nest points.

We calculated the loss of suitable habitat for the giant ibis over time. During the wet season, 75% of nest points were found over surface water, while 75% of non-nest points were found within 14.5m of surface water (Fig 5A). During the dry season 75% of nest points were found within 92.7m from surface water and 152.7m for non-nest points (Fig 5B). We found a decline in the median areas of suitable habitat of 1791km$^2$, equivalent to a 25% loss of suitable area

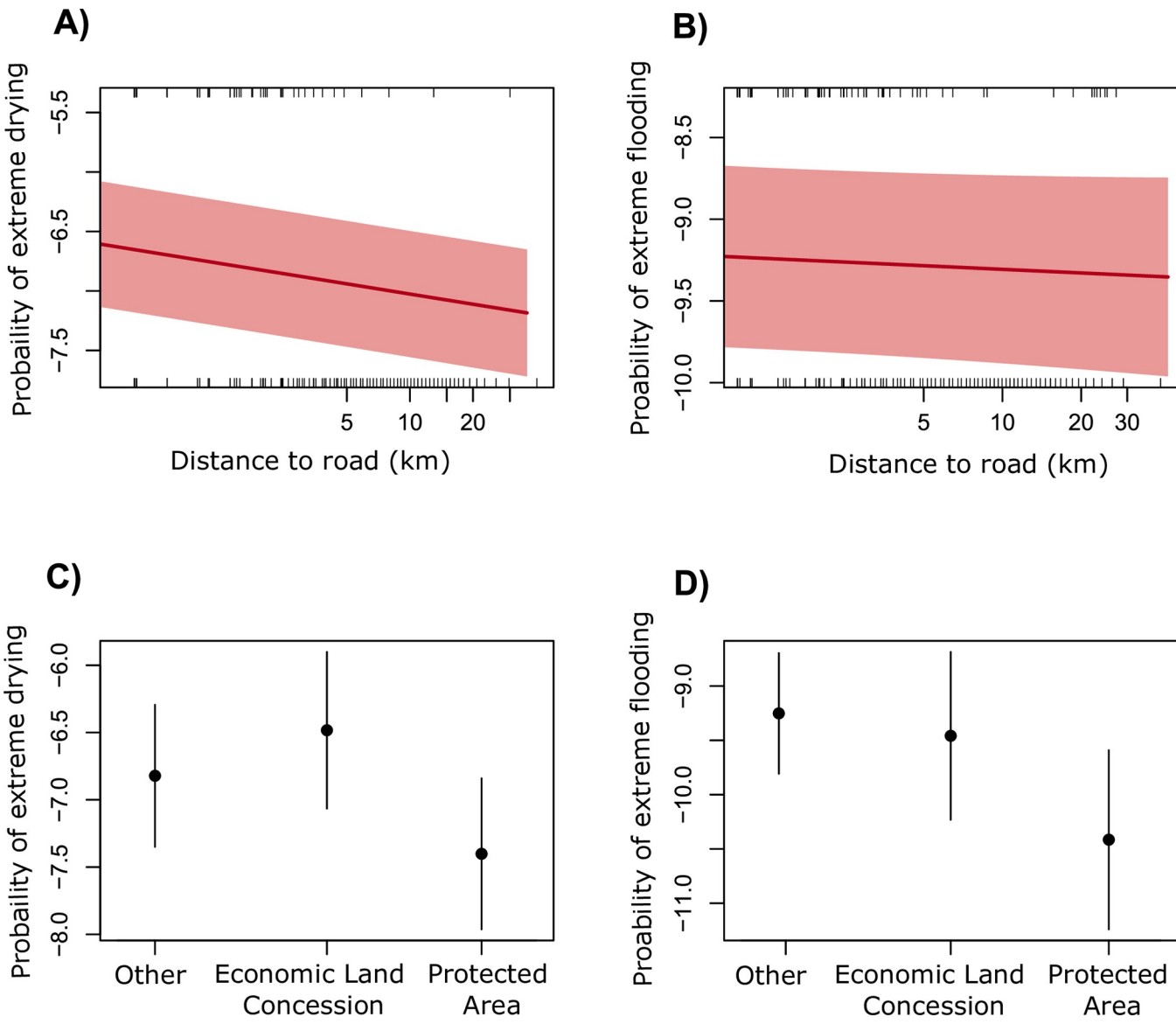

**Fig 4. Effect plots for binary INLA models showing the logit probability.** Plots to show the logit probability of extreme land drying (A) and extreme flooding (B) in relation to distance of pixels to roads. Plotted is the mean (0.5) with 95% credible intervals. The marginal rug plots display the distribution of the data for pixels experiencing (top) or not experiencing (bottom) large changes in surface water over time. Forest plots show the median (point) and the 25th and 75th percentiles of the logit probability of extreme land drying (C) and flooding (D) within other land uses, Economic Land Concessions (ELCs) and protected areas (PAs).

during the dry season (Fig 6). We also found a decline in the median area of surface water during the wet season of 1718km$^2$ which equates to a 21% reduction (Fig 6).

## Discussion

Our study uses remote sensing to determine the change in availability of surface water, joining a growing body of work [22, 49]. We highlight the value of these methods not only to quantify changes in surface water, but also to help identify drivers meaning our results have direct application for the conservation of species within this region. Our results confirmed our first hypothesis and showed a loss in surface water availability over time, which is in line with global

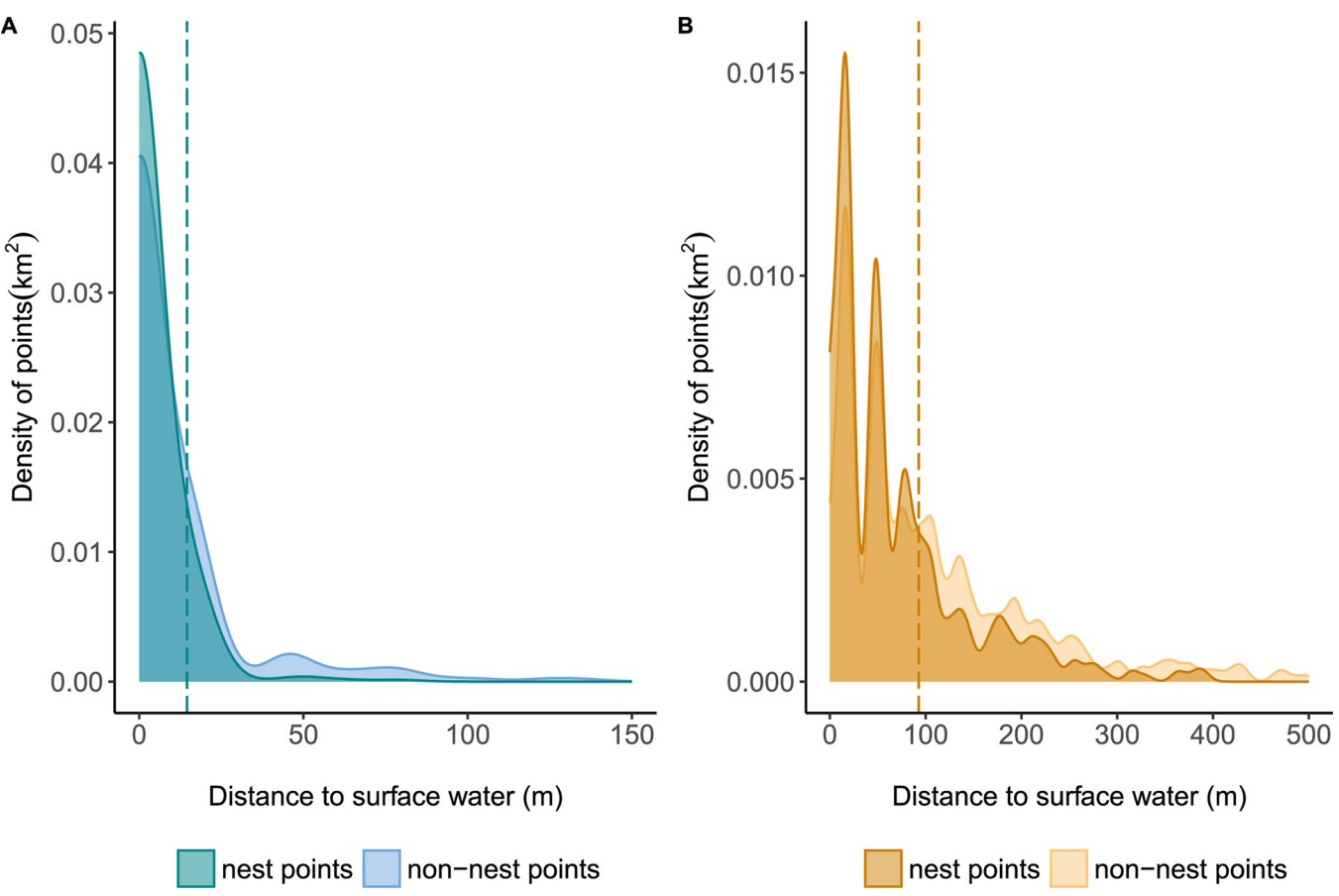

**Fig 5. Density plots comparing the distance to water of nest and non-nest points.** Plots to show the distance of nest and non-nest points to surface water during the month when the nest was recorded (A) and the distance to surface water, of the same points, during the dry season only. Vertical lines indicate the 75th percentile of distances used to define suitable giant ibis habitat.

trends [15]. We found the driving force of this was the extensive change to land use, specifically the presence of Economic Land Concessions and roads which is also supported by other studies [50–52]. ELCs are extensive areas of monoculture cash crops that cause removal of natural habitats and have been shown to affect local hydrological processes such as runoff [52, 53]. For example, in a 2011 survey of waterholes in in protected areas of the Northern Plains of Cambodia 40% of waterholes has been lost due to land clearance for ELCs [54]. Cash crops such as rubber and sugarcane, are also very water intensive which puts a lot of strain on water resources [11]. ELCs are extensive in Cambodia, covering 2.3 million hectares and the damage is long-term as the land is leased from 70 up to 99 years [51, 55]. Cambodia has one of the highest rates of deforestation globally, and the main driver is agricultural expansion (much of this for ELCs), which has increased from 1% (1997) to 61% (2016) [56]. Roads, as this study found, have a negative impact on surface water, and as other studies have shown, they facilitate the removal and fragmentation of habitats and destruction or deviation of water courses [57, 58]. They can disrupt floodplains, impacting water flow, sediments, nutrients and aquatic life, having a negative impact on hydrology [50]. Land use is causing huge changes in surface water availability and while we have not considered HEP dams directly, which is a limitation to this study, other research has found that it has had significant impacts within the Mekong basin [14, 15, 59]. There are 28 large dams in the Mekong basin itself and plans to develop a further

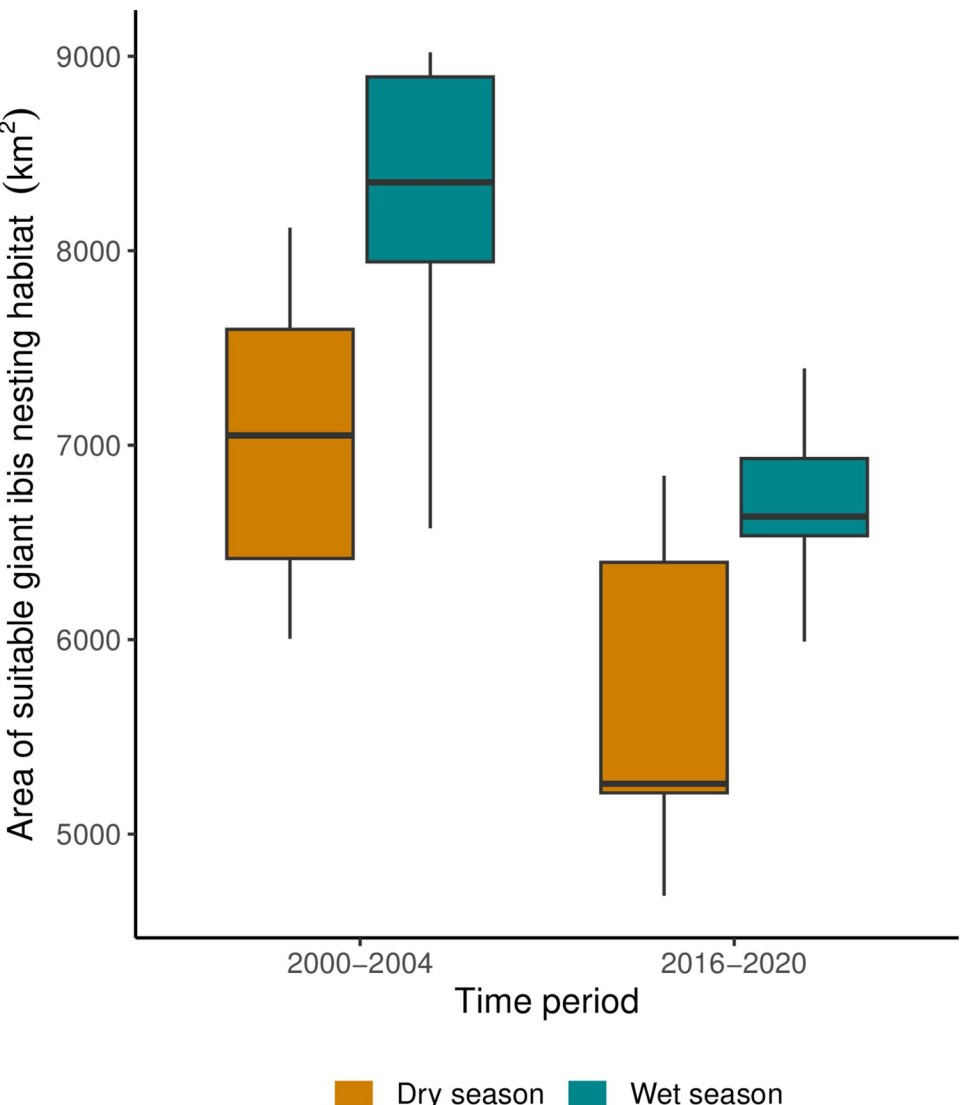

**Fig 6. Boxplot to show the availability of giant ibis habitat.** The median (line) and the 25th and 75th percentiles of available area of suitable giant ibis habitat are plotted (pixels within 75% of recorded distances to water for real nest locations) to compare the availability between the first (2000–2004) and second (2016–2020) time periods.

11 in the mainstream of the Mekong in Laos and Cambodia, meaning impacts on flow are wide reaching and likely to increase in the future [60]. Dams can have varied effects on river flow, such as increases recorded for upstream flow causing bank erosion, permanent inundation of habitats and reduction in maximum downstream flow [15]. Dams also impact local communities, water levels, sediment transport, fish migration and reduce water quality, affecting the habitats, species and the 65 million people living within the Mekong Basin [15, 61].

Our study found no correlation between the availability of surface water and precipitation, despite declines in mean annual precipitation over time. This has also been shown by other authors, who found a -0.184% change in rainfall across Cambodia per year between 1951–2002 [25]. However, despite the lack of correlation found within this study, climate change is predicted to have significant impacts on the Mekong Basin [25, 62]. By 2100 there is expected to be a 1–8% increase in precipitation meaning that flooding and discharge during the wet

season will increase causing inundation of key ecological habitats and the loss of 75% of current floodplains [17]. Extended drought during the dry season is also expected leading to increased risks of water shortages, land degradation and desertification affecting all life within the basin [16]. Our study suggests that local land use conversion is driving the changes to a higher degree than the wider-scale precipitation patterns, but climate change will undoubtedly amplify the impacts of this in the future. Such a pattern has also been evidenced in other literature, highlighting the influence of land use change on surface water availability. Before the construction of the first major Mekong dam, climate change was linked to 82% of the flow change (1991–2009) [15]. In contrast, post major dam construction (2010–2014) 62% of changes to flow were linked to HEP which shows the overwhelming influence of changing land use on surface water [15].

Land use change is one of the main threats to natural habitats globally and one way to prevent this is the managing land in a way that supports biodiversity. This is particularly pertinent as inland freshwater tends to be underrepresented in protected areas globally [63]. We found that protected areas within Cambodia had a lower likelihood of extreme flooding or drying events, surface water conditions appeared to be more stable over time. This likely reflects more stable land use within protected areas, perhaps due to lower levels of land clearance due to law enforcement. Protected areas cover 6.3million ha (2021) across Cambodia and as one study has shown can have a positive impact on forest habitat protection [64]. Despite this, protected areas continue to be under threat and are still targeted for exploitation, especially those bordering or overlapping ELCs [64–66]. 70% of ELCs designated in Cambodia by 2012 were within protected areas providing an ongoing existential threat to apparently protected areas [55]. The full implementation of the 2012 moratorium placed on the creation of new ELCs in Cambodia should also help to control future widespread habitat clearance for ELCs [67].

Our results show how the loss of surface water will affect the Critically Endangered giant ibis. We found that while small in absolute terms, the overall decline in surface water availability translates to a disproportionately large loss in suitable habitat for the giant ibis. This decline has negative implications for a species inextricably linked to surface water for its forage and already confined to protected areas in small and declining populations [35, 68]. The disproportionate impact of reduction in availability of surface water has widespread impacts, likely affecting many other species such as banteng *Bos javanicus*, Eld's deer *Rucervus eldii* and white-shouldered ibis *Pseudibis davisoni* that also rely on availability of surface water within this region [20]. During the dry season these species struggle to find reliable water sources, limiting their distribution and foraging opportunities [27], which makes understanding the changes in surface water even more vital.

## Conclusions and recommendations

To conclude, we found a decline in surface water across our study site, predominantly driven by land use change. While small overall, the impact of this loss is amplified for species such as the giant ibis that rely on sites near water. Therefore, our main recommendation is to promote better protection of this vital resource. For this we need to understand hydrology of seasonal wetlands more specifically, and we recommend pairing remote sensing with on the ground water source monitoring to also understand the fine scale nuance to apply the most effective management. Practical management, such as restoration of temporary water sources, can be informed by increased understanding of local threats, physical characteristics and life cycles of individual sources such as waterholes. Waterhole restoration would be an effective practical step to tackling the declines found within this study and to support wider conservation goals in Cambodia. This understanding is particularly vital due to the implications for people living

within these systems. Their reliance on this diminishing resource needs to be highlighted and steps taken towards management of surface water and prevention of further declines to ensure human wellbeing.

## Supporting information

**S1 Fig. Map of economic land concessions within the study area.** This map shows the location of the Economic Land Concessions (ELCs, highlighted in red) present within the Study Site which is the giant ibis estimated distribution (yellow envelope). The protected areas (PAs) are also shown (grey) to give spatial context of the ELCs in relation to Cambodia's PAs. Protected areas, Economic Land Concessions and water bodies data included in this figure has been published by Open Development Cambodia herein are licensed under a CC BY-SA 4.0. The giant ibis distribution data included in this figure has been reprinted from [34] under a CC BY licence, with permission from BirdLife International, original copyright [2019].
(DOCX)

**S2 Fig. Map to show the location of giant ibis nest sites.** This map shows the location and distribution of the 438 giant ibis nests (black triangles), recorded between 2003 and 2020 within the northern plains protected areas: A) Kulen Promtep Wildlife Sanctuary, B) Prey Preah Roka Wildlife Sanctuary and C) Chhaeb Wildlife Sanctuary. The nest location data has been plotted over the surface water transition map to show changes in surface water between 2000–2004 and 2016–2020.
(DOCX)

**S3 Fig. Figure to show trend analysis results for mean annual precipitation.**
(DOCX)

**S1 Text. Additional information on the covariate data used within the analysis.**
(DOCX)

**S2 Text. Field methodology for giant ibis nest monitoring.**
(DOCX)

**S1 Table. Table of surface water transition categories.** The table below summarises the 16 transition categories that we used to define the pixel transitions. The transitions categories for the first (2000–2004) and last periods (2016–2020) were defined. The transition value denotes the number of categories that the pixel has changed by between the two periods. QGIS was used to calculate the percentage area of each surface water state and transition category to reflect the extent of the change.
(DOCX)

**S2 Table. Table to summarise the sensitivity checking results for the giant ibis habitat availability analysis.** This table summarises the percentile value comparison carried out for sensitivity checking for the giant ibis suitable habitat area. We calculated the 65th, 75th and 85th percentiles of the data for comparison. We chose the 75th percentile to define the area that we considered as suitable habitat for the giant ibis. There was little difference between the 65th and 85th percentiles when compared with the 75th percentile values so the 75th was selected. We used this value to determine the area of suitable giant ibis habitat lost between the two time periods in both the wet and dry season.
(DOCX)

**S3 Table. Tables to show trend analysis results for mean annual precipitation.** Table to summarise the linear model results conducted for the mean annual precipitation over time.

We created a linear model to explore the trend in precipitation over time.
(DOCX)

**S4 Table. Table to summarise the Pearsons correlation test results.** Table to summarise the Pearson correlation test results for the relationship between surface water availability and precipitation and dry water availability during the dry season and precipitation. We found no correlation for either of the tests.
(DOCX)

**S5 Table. Tables of results from the INLA models.** These tables summarise the results from A) Model 1 which tested for general changes (drying and flooding) across the study site; B) Model 2 which tested for the probability of extreme drying events which are defined as those pixels that transitioned by two or more categories drier; and C) Model 3 which tested for the probability of extreme flooding events which are defined as those pixels that transitioned by two or more categories wetter. It displays the effect of the selected covariates on the surface water within the study site using the gaussian general model. Here we have shown the mean values, the upper and lower quantile ranges. The bold values identify those covariates that have a significant effect on the surface water (have ranges that do not overlap zero).
(DOCX)

**S6 Table. Table of results for comparison of distance from nest and non-nest points to surface water.** Table to show the results from the Mann-Whitney U test comparing the distance to surface water from nest and non-nest points. We tested this relationship during both the dry and wet seasons. In both seasons nest points were found significantly closer to areas of surface water than non-nest points.
(DOCX)

## Acknowledgments

The authors would like to thank the Wildlife Conservation Society Cambodia for their provision of knowledge, data and support. In particular the WCS/ USAID GPL field team, led by Mao Khean, who collected the giant ibis nest data. The following people were involved in the data collection; Chan Vay, Chhieng Chhie, Chhit Ret, Choeun Sothi, Choeun Y, Chunn Samroen, Doung Ri, Hiem Kimheng, Khoeun Sokha, Leam Minea, Lom Vanna, Loun Voleak, Mao Khean, Mok Pich, Mom Set, Mom Sokheourng, Nann Boarn, Peak Saven, Phay Chandaro, Proeun Heng, Prom Vorn, Pun Phoeung, Sean Sokny, Soeun Sorn, Soeun Sen, Tan Sophan, Tem Hoeung, Thoeun Thieng, Touch Doeu and Vann Meat. We thank the Cambodian Ministry of Environment for enabling the completion of this work under the framework of a memorandum of understanding between the Wildlife Conservation Society and themselves. There is no specific permit number for this data collection as it is included within the annual work plan created by the two parties. We also thank Colin McClean for his valuable feedback on early drafts of this paper.

## Author Contributions

**Conceptualization:** Louisa Mamalis, Colin M. Beale.

**Formal analysis:** Louisa Mamalis, Colin M. Beale.

**Investigation:** Louisa Mamalis, Mao Khean.

**Methodology:** Colin M. Beale.

**Software:** Louisa Mamalis, Colin M. Beale.

**Supervision:** Kathryn E. Arnold, Simon P. Mahood, Colin M. Beale.

**Visualization:** Louisa Mamalis.

**Writing – original draft:** Louisa Mamalis.

**Writing – review & editing:** Kathryn E. Arnold, Simon P. Mahood, Mao Khean, Colin M. Beale.

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
