## [Decision Letter · Decision Letter 0]

9 Apr 2024

PONE-D-24-10904Quantifying the availability of seasonal surface water and identifying the drivers of change within tropical forests in CambodiaPLOS ONE

Dear Dr. Mamalis,

Thank you for submitting your manuscript to PLOS ONE. After careful consideration, we feel that it has merit but does not fully meet PLOS ONE’s publication criteria as it currently stands. Therefore, we invite you to submit a revised version of the manuscript that addresses the points raised during the review process.

We look forward to receiving your revised manuscript.

Kind regards,

Bijeesh Kozhikkodan Veettil

Academic Editor

PLOS ONE

Journal Requirements:

"This work was funded by the Natural Environment Research Council, UK Research Institute under the Adapting to the Challenges of a Changing Environment DTP https://accedtp.ac.uk/. The grant was awarded to LM to complete this work [grant code: NE/L002450/1, 2020]."

"..Funding: This work was supported by the Natural Environment Research Council, UK Research Institute under the Adapting to the Challenges of a Changing Environment DTP [grant code: NE/L002450/1, 2020]."

Please remove any funding-related text from the manuscript.

5. We note that you have indicated that there are restrictions to data sharing for this study. PLOS only allows data to be available upon request if there are legal or ethical restrictions on sharing data publicly. For more information on unacceptable data access restrictions, please see http://journals.plos.org/plosone/s/data-availability#loc-unacceptable-data-access-restrictions. 

7. We note that Figures 1, 3 and Supporting Figure (S6 Fig) in your submission contain map images which may be copyrighted. All PLOS content is published under the Creative Commons Attribution License (CC BY 4.0), which means that the manuscript, images, and Supporting Information files will be freely available online, and any third party is permitted to access, download, copy, distribute, and use these materials in any way, even commercially, with proper attribution. For these reasons, we cannot publish previously copyrighted maps or satellite images created using proprietary data, such as Google software (Google Maps, Street View, and Earth). For more information, see our copyright guidelines: http://journals.plos.org/plosone/s/licenses-and-copyright.

1) You may seek permission from the original copyright holder of Figures 1, 3 and Supporting Figure (S6 Fig) to publish the content specifically under the CC BY 4.0 license.  

2) If you are unable to obtain permission from the original copyright holder to publish these figures under the CC BY 4.0 license or if the copyright holder’s requirements are incompatible with the CC BY 4.0 license, please either i) remove the figure or ii) supply a replacement figure that complies with the CC BY 4.0 license. Please check copyright information on all replacement figures and update the figure caption with source information. If applicable, please specify in the figure caption text when a figure is similar but not identical to the original image and is therefore for illustrative purposes only.

Reviewers' comments:

Reviewer's Responses to Questions

**Comments to the Author**

1. Is the manuscript technically sound, and do the data support the conclusions?

Reviewer #1: Yes

Reviewer #2: Partly

2. Has the statistical analysis been performed appropriately and rigorously? 

Reviewer #1: Yes

Reviewer #2: I Don't Know

3. Have the authors made all data underlying the findings in their manuscript fully available?

Reviewer #1: Yes

Reviewer #2: Yes

4. Is the manuscript presented in an intelligible fashion and written in standard English?

Reviewer #1: Yes

Reviewer #2: Yes

5. Review Comments to the Author

Reviewer #1: Reviewer Comments

Introduction

• The introduction provides a comprehensive overview of the importance of surface water and the threats it faces globally. However, it could benefit with a more detailed and articulate explanation of the study's objectives in the context of broader literature.

Materials and Methods

1. Study Area:

The description of the study area is good and detailed. However, the author should provide more details about the climate of the study area, for example, different graphs and values on the average monthly and yearly rainfall, temperature, and humidity. Since rainfall is an important part of the study, different graphs of the historical average monthly, yearly, and seasonal rainfall would be beneficial.

2. Data Analysis:

The authors have not mentioned if any other image pre-processing is carried out besides cloud masking. It would be beneficial to mention if any other image pre-processing was carried out.

The authors should also mention why Landsat was used, instead of other sources of satellite images, like Sentient 2.

3. Implications for the giant ibis:

The authors should consider providing more explanation on how the spatial regression approach using INLA modeling accounts for spatial autocorrelation and the choice of covariates in the model.

Results

• It would be recommended for the authors to carry out a trend analysis of precipitation, to see if there is any trend in precipitation if it is decreasing, and how this could be affecting the surface waters.

• Additionally, it is recommended the authors provide more detail on the statistical methods used to analyze the data, including any assumptions made and how they were validated.

Discussion

• It is recommended that the authors reference similar studies and make comparisons with similar studies.

• Also, the authors should provide the uncertainties and limitations of the study.

• Additionally, the authors should provide a conclusion section, providing a summary of key findings and including recommendations for future studies, including how the findings could inform conservation strategies or management practices.

Overall Comment

Overall, the paper is well-written and can be accepted for publishing after making a few changes. Importantly the authors need to give more details in the Study area and methods section and give some reference to other studies in the results and discussion. Additionally, the authors need to include conclusion and recommendations sections.

Reviewer #2: The general approach of the paper is very interesting and well thought. However, I would need to be able to see the code of the analysis to recommend acceptance. Without google earth engine, this seems not possible. I would therefore recommend that the authors publish the code in an open access repository.

6. PLOS authors have the option to publish the peer review history of their article (what does this mean?). If published, this will include your full peer review and any attached files.

Reviewer #1: No

Reviewer #2: No

---

## [Author Response · Author response to Decision Letter 0]

18 Jun 2024

Dear Dr. Kozhikkodan Veettil,

Many thanks for your response to the submission of the paper titled ‘Quantifying the availability of seasonal surface water and identifying the drivers of change within tropical forests in Cambodia’, submission number: PONE-D-24-10904. I really appreciate you and the reviewers taking the time to evaluate this research and provide some useful feedback to improve it. 

Please find a table below with information on how we have addressed and responded to each comment within the review. 

Comment Response

Editor comments:

1.Please ensure that your manuscript meets PLOS ONE's style requirements, including those for file naming.

Completed: I have referred to the PLOS ONE formatting guidelines and have edited to my manuscript accordingly. 

Completed: There is no specific permit number for the data collection associated with this study. I have therefore included a statement to explain this within the acknowledgements section. 

Completed: I have added in the sentence regarding funder involvement to my revised cover letter. 

4. Please note that funding information should not appear in the Acknowledgments section or other areas of your manuscript. We will only publish funding information present in the Funding Statement section of the online submission form. Please remove any funding-related text from the manuscript.

Completed: I have removed any reference to the funders from my acknowledgements section. 

5. We note that you have indicated that there are restrictions to data sharing for this study. PLOS only allows data to be available upon request if there are legal or ethical restrictions on sharing data publicly.

Completed: I have added information to the data availability statement explaining why we are unable to share the giant ibis nest location data for ethical reasons. 

Completed: I have created a github repository where all the scripts are saved and available for this project. 

Completed: I have written up the Google Earth Engine scripts as .txt files so that they are accessible to anyone to read via the github repository. 

Completed: I have saved and commented on the relevant R scripts for reference. These are also available via the gitbuh repository. 

Completed: I have updated the access to the colab analysis script to ‘share with anyone with the link’. I have also updated the comments on this script as well. This remains available through the url in the data availability statement. 

7. We note that Figures 1, 3 and Supporting Figure (S6 Fig) in your submission contain map images which may be copyrighted. All PLOS content is published under the Creative Commons Attribution License (CC BY 4.0), which means that the manuscript, images, and Supporting Information files will be freely available online, and any third party is permitted to access, download, copy, distribute, and use these materials in any way, even commercially, with proper attribution. For these reasons, we cannot publish previously copyrighted maps or satellite images created using proprietary data, such as Google software (Google Maps, Street View, and Earth). We require you to either (1) present written permission from the copyright holder to publish these figures specifically under the CC BY 4.0 license, or (2) remove the figures from your submission.

Completed: I have added in the license information into the figure captions for the data sources used to create Fig 1, Fig 3 and S6 Fig. I have attached the signed permission to publish form completed by myself and BirdLife to my resubmission. I have also added in this information to the data availability statement. I have also edited Fig 1 slightly by adding the location of Economic Land Concessions to the map. I have put it through PACE for checking and have uploaded this edited figure. 

8. Please include captions for your Supporting Information files at the end of your manuscript, and update any in-text citations to match accordingly. 

Completed: I have added all the titles and captions of my Supporting Information at the end of the main text. I have updated these in the Supporting Information documents as well. Each Supporting Information has been saved as a separate file. 

Completed: I have reviewed all my references to ensure they are all correct and included within the text. I have also converted all my references into the correct PLOS ONE format. 

Reviewer #1 comments

1.Introduction:

The introduction provides a comprehensive overview of the importance of surface water and the threats it faces globally. However, it could benefit with a more detailed and articulate explanation of the study's objectives in the context of broader literature.

Completed: I have added in references to broader literature to provide more context within the introduction and to emphasise why this study’s objectives. 

2.Study Area:

The description of the study area is good and detailed. However, the author should provide more details about the climate of the study area, for example, different graphs and values on the average monthly and yearly rainfall, temperature, and humidity. Since rainfall is an important part of the study, different graphs of the historical average monthly, yearly, and seasonal rainfall would be beneficial.

Completed: I have added in some relevant information regarding the climate in Cambodia, specifically referencing Thoeun et al., 2015 which has detailed information on Cambodia’s climate, including a number of historical plots. 

3.Data Analysis:

The authors have not mentioned if any other image pre-processing is carried out besides cloud masking. It would be beneficial to mention if any other image pre-processing was carried out. The authors should also mention why Landsat was used, instead of other sources of satellite images, like Sentient 2.

Completed: I have added in more details regarding the pre-processing of the LandSat 7 data, including more elaboration on the cloud masking and the removal of a scanline error within the data.

Completed: I have added information into the methods section explaining why we chose to LandSat 7 satellite images for this study. 

4.Results:

It would be recommended for the authors to carry out a trend analysis of precipitation, to see if there is any trend in precipitation if it is decreasing, and how this could be affecting the surface waters. Additionally, it is recommended the authors provide more detail on the statistical methods used to analyze the data, including any assumptions made and how they were validated.

Completed: I have completed a linear regression to look at the trend in precipitation over time. I have added the statistical results into the results section and have added in a figure and table outlining the findings in the Supporting Information (S6 Fig and S7 Table).

Completed: I have added in more information about how the models worked, including assumptions that we made and how we validated them. 

5.Discussion:

It is recommended that the authors reference similar studies and make comparisons with similar studies. Also, the authors should provide the uncertainties and limitations of the study. Additionally, the authors should provide a conclusion section, providing a summary of key findings and including recommendations for future studies, including how the findings could inform conservation strategies or management practices.

Completed: I have added in a lot more references to similar studies and direct comparisons to this literature within the discussion.

Completed: I have included information on limitations to this study, such as not considering HEP dam influence directly within the analysis and the use of remote sensing alone with no ground truthing data.

Completed: I have added in a conclusion and recommendations section to the end of paper which summarises the key findings as well as the recommendations for future work and how the results can inform conservation strategies in this area. 

Reviewer #2 comments

The general approach of the paper is very interesting and well thought. However, I would need to be able to see the code of the analysis to recommend acceptance. Without google earth engine, this seems not possible. I would therefore recommend that the authors publish the code in an open access repository.

Completed: I have cleaned and commented all of the scripts that were used for this analysis and made them available in a git hub repository. 

I hope this information is enough to address the changes suggested for this manuscript. Should you require any more changes or additional information then please do not hesitate to contact me. 

Sincerely 

Louisa Mamalis

PhD researcher 

Department of Biology 

University of York

York

---

## [Decision Letter · Decision Letter 1]

16 Jul 2024

Quantifying the availability of seasonal surface water and identifying the drivers of change within tropical forests in Cambodia

PONE-D-24-10904R1

Dear Dr. Mamalis,

We’re pleased to inform you that your manuscript has been judged scientifically suitable for publication and will be formally accepted for publication once it meets all outstanding technical requirements.

Kind regards,

Bijeesh Kozhikkodan Veettil

Academic Editor

PLOS ONE

Additional Editor Comments (optional):

Reviewers' comments:

Reviewer's Responses to Questions

**Comments to the Author**

1. If the authors have adequately addressed your comments raised in a previous round of review and you feel that this manuscript is now acceptable for publication, you may indicate that here to bypass the “Comments to the Author” section, enter your conflict of interest statement in the “Confidential to Editor” section, and submit your "Accept" recommendation.

Reviewer #1: All comments have been addressed

2. Is the manuscript technically sound, and do the data support the conclusions?

Reviewer #1: Yes

3. Has the statistical analysis been performed appropriately and rigorously? 

Reviewer #1: Yes

4. Have the authors made all data underlying the findings in their manuscript fully available?

Reviewer #1: Yes

5. Is the manuscript presented in an intelligible fashion and written in standard English?

Reviewer #1: Yes

6. Review Comments to the Author

Reviewer #1: (No Response)

7. PLOS authors have the option to publish the peer review history of their article (what does this mean?). If published, this will include your full peer review and any attached files.

Reviewer #1: No

---

## [Editor Report · Acceptance letter]

19 Jul 2024

PONE-D-24-10904R1 

PLOS ONE

Dear Dr. Mamalis, 

I'm pleased to inform you that your manuscript has been deemed suitable for publication in PLOS ONE. Congratulations! Your manuscript is now being handed over to our production team.

Kind regards, 

on behalf of

Dr. Bijeesh Kozhikkodan Veettil 

Academic Editor

PLOS ONE